# OpenReview forum: "Uncoupled and Convergent Learning in Monotone Games under Bandit Feedback"
_NeurIPS.cc/2025/Conference — NeurIPS 2025 poster_

### Official Review · Reviewer_8yuX · 2025-06-26

**Clarity:** 3
**Significance:** 3
**Originality:** 3
**Rating:** 4
**Confidence:** 5

**Summary:**

The article presents a set of mirror descent algorithms for oblivious learners in continuous action monotone games. These learners only take actions and receive rewards - they do not receive gradient information or any information about actions or rewards of other players. The authors consider each of static games, converging games, and drifting games, giving state of the art convergence guarantees in each case, in terms of the last iterate convergence. A customary cursory experimental demonstration is provided.

**Questions:**

Lines 152-158, and Lines 193-195. Please provide some discussion at least flagging up the issue of how to choose p when learners are supposed to be quite oblivious. Ideally some realistice examples would be given where the reader could see how p could be chosen (I don't count c(x)=x^2).

Line 155: “convex but not linear”. Do we just mean “strictly convex”? I’m concerned that a function c_i that is linear in part of the function space but strictly convex elsewhere might not admit a kappa and admissible p while staying convex.

Lines 160-168 repeat the conditions on h twice over.

Line 174. Previous works have had to work to ensure the actually-played \hat{x}_i^t are in the feasible set. I suspect that the ellipsoidal estimator, combined with the barrier function, does this automatically? Please can you just note this fact - it is not clear from what is presented in the article.

Algorithm 1, Inputs line. Have I missed something? There a constant kappa that has appeared out of nowhere. What is this and what effect does it have on anything?

Algorithm 1, line 4. This would read much more nicely if it started with "sample z", before then defining \hat{x}, then receiving the feedback (ie the order in which it happens in the code).

Line 207, and other similar theorem statements. The convergence is given in terms of the sum of the Bregman divergences with respect to the game-dependent, user-chosen p regularizer. In contrast, most results in the literature that are given with respect to Bregman divergences are with respect to a non-game-dependent regularizer such as the h in this article. Some discussion of this would be valuable.

Line 214, the term "a fixed point given" is confusing - it could be interpreted that omega is a fixed point of the dynamic (which it is not). I think "omega_i is any strategy" would suffice?

Section 5 generally. If mu = 0 it is far from clear to me that there is a unique Nash equilibrium. Yet your results implicitly assume there is. Please can you clarify or fix this.

Line 228. “While it does not imply a bounded first term”. The first term in question is a Bregman divergence with respect to barrier regulariser h. The only way the first term could be “unbounded” for fixed \omega is if \omega was on the boundary. Since you want \omega to be a Nash equilibrium, you ought to say when a Nash equilibrium is on the boundary, \omega has to be carefully chosen. Boundary Nash are often distinguished in the literature but you make no mention of this.

Theorems 5.2 and 5.3. kappa does not appear in these results as stated. It vanishes some way through the proofs. You need to decide if it should be dropped from Theorem 5.1, or included in these two theorems.

Line 271. I think the only change to algorithm 1 is to replace kappa with tau. And now we specify tau. If I am correct, then this feels like a peculiar way to present it. Why not just specify that now you take kappa=1/T^{1/6} and carry on. Also, within this theorem, you do not specify what delta_t should be. And your results are in terms of G, whereas you have defined G_p in the first line of the statement.

Lines 295-297. These appear to claim both that Duvocelle provided a result and that you are the first to provide the same result. Please clarify.

Line 300. By writing down x_i^{t,*} you once again assume a unique equilibrium at each time point. I don't believe conditions have been specified which ensure the uniques of the equilibrium. Please clarify.

Line 303. The key innovation of the paper, the inclusion of the D_p regularizer, is now not included in the the mirror map. Is that deliberate? Please explain why omitting this is now the right thing to do.

p9, some better discussion of the results would be extremely helpful. In particular, are the plots showing what we would expect to see (i.e. what is the expected outcome of each of the competitor algorithms, and is this what we see). Is the higher initial variance from your algorithm to be expected? Is the non-convergence of GD with estimate expected? Being consistent with your color scheme across the plots would be really helpful too - green is GD with gradient on the outside plots, but GD with estimated gradient on the middle plot...

Lines 324 and 325 seem to contradict line 312?

I have not read all of the proofs in great detail. They look to be about right, but there is a lot of algebra and a lot of repetition. They are not particularly nicely presented - the reader has to do a lot of their own work to follow from one line to the next. I did notice the following problems:

Line 552: "Similar to Theorem 5.1" is a little too much of a leap for me. Theorem 5.1 starts by referring directly to Lemma J.1. Lemma J.1 is entirely based on fixed cost functions. I expect that if I were to replace the c's in Lemma J.1 with c_t's then the difference would be the B\sum\Delta^t term. But a reader should not have to work through the proof of Lemma J.1 to be confident that all the other terms (except the last 2) in Lemma J.1 are not dependent on c being fixed. I would have thought a Lemma J.1 in terms of possibly evolving cost functions, that could then be specialized by fixing them, would be more useful.

Line 614. Lemma J.3 is for G-\nabla p being monotone. But here you state (essentially) that G-p is monotone. I believe that, since p is strictly convex, the result may hold anyway. But as written the logic is incorrect I think. Note also that Lemma J.3 is not quite what is needed - it has a single term on the LHS whereas this result has a sum on the LHS (see also comments on Lemma J.3).

Line 615. You should really offer some more help to the reader than “Therefore”. Perhaps “Summing  [Eqn no for line 613] and using [eqn no for line 614] to replace the third term on the right hand side, we can then drop the non-negative D_p(x,omega) term to get” or something. Of course, dropping out the D_p from the inequality in Line 614 (while explaining that you do so) would mean less going on here.

Line 616. Technically, you are summing over t, not over T. There’s a lot going on here too – these three lines took me half an hour to work out! Some more help to the reader would be very valuable. Perhaps “Summing over T, cancelling both sum_{t,i} D_h and sum_{t,i} tD_p terms, and discarding the non-negative D_h(omega,x^{T+1}, we have”. You’re also missing an eta_1 multiplier on the second term of the RHS.

Line 621, and Lemma J.6, need to be significantly clearer. In the lemma, pi is undefined, as is {\cal X}_\epsilon. Both need to be defined. Then Line 621 needs to justify why the claimed result is a consequence of the lemma. Using the same words as the lemma would help (i.e. I think that x_i^1 is a center and I think it is implicitly claiming that omega\in {\cal X}_{1-1/\sqrt{T}}, but it is not clear how this fits together or why we have a bound of nu\log(T).

Line 622, last two sub-lines, you have sum eta_t terms outside of the sum over t, which I think should be inside.

Line 629. I think, since f changes with both x_i^t and t you should probably include these as arguments?

Lines 630-631. This needs a lot more explanation. I don’t believe it has been stated that D_p or D_h are self-concordant barriers, so why does adding the linear term make f a self-concordant barrier?

Line 632. This should be an inequality, not an equality.

Line 634 there is a mistake. A_i^t is defined on p5 to be (\nabla^2 f(x_i^t))^{-1/2}, but here it is taken to be (\nabla^2 f(x_i^t))^{-1}. Additionally, the second line of the equations is, I think, never used?

Line 635. You take x_i^{t+1} to be argmin f(x_i^t). But it should be argmin_{x_i} f_{x_i^t,t}(x_i) (i.e. the x_i^t and t are fixed arguments, whereas it’s the argument x_i of f that you maximise over). Also, there is a mistake here – the kappa term from algorithm 1 is not part of f, so this armin is not correct.

Line 639. Is p a function from the combined action space X, or from individual action spaces X_i? The first line here implies the former, but the second the latter. There’s also, within the lemma, no obvious reason why the nabla_p (x) – nabla_p(x’) relates to a sum over i of different D_p(x_i,x_i’) terms. I would advise simply making the lemma more specific to the use case in the article.

**Ethical Concerns:**

["NO or VERY MINOR ethics concerns only"]

**Final Justification:**

I think the work is nice. There is a weakness in that choosing p_i for an oblivious player is hard, and it's not obvious how to do it. There's also a weakness that the convergence result depends on the choice of p. The authors have committed to being explicit about this in the final article. So I think the article is good enough to publish, but the weaknesses ensure it is only a weak accept.

**Limitations:**

The article should definitely discuss the limitations from needing to know the cost functions well enough to be able to select an appropriate p. Other than that, I think any limitations are well specified in the results.

**Quality:**

3

**Strengths And Weaknesses:**

The article achieves better results than have yet been published, with a O(1/T^0.25) results in monotone games, improving to O(1/T^0.5) for strictly monotone games, with the obvious modifications for converging and drifting games. These are strong results, showing a high quality paper.
There is a lot squeezed in to one paper here. This is both a strength and a weakness. It means there is a lot of progress, but is quite difficult to follow all that is attempted to be presented, even for a somewhat expert reader (I have published closely related papers and currently supervise PhD students on this topic). Clarity is somewhat weak as a result.
While the algorithm is somewhat standard, the key innovation which allows these new results is the introduction of the assumption in lines 152-158. Two different regularizers allows better control of the learning rates. However this also brings a weakness - it is far from clear how an oblivious learning could choose there p regularizer, since it depends quite strongly on the particular form of the game's cost functions.

---

> ### Author Rebuttal · Authors · 2025-07-30
>
> We thank the reviewer for the detailed and constructive comments.
>
> ### General Acknowledgment on Proof Comments:
>
> We sincerely thank the reviewer for their careful and detailed reading of the proofs. Their understanding is correct, and we very much value their comments and suggestions for improving the clarity and accessibility of the arguments. We will revise the manuscript to address all of the issues raised, including improving transitions between lines, clarifying technical assumptions (e.g., monotonicity and the structure of evolving cost functions), fixing notation inconsistencies, and expanding explanations where necessary to assist the reader in following the derivations.
>
> ### On the uniqueness of the Nash equilibrium:
>
> We appreciate the reviewer’s careful attention to this point. We do not assume the Nash equilibrium is unique. Our results, including Theorem 5.1, are stated to guarantee convergence to some Nash equilibrium, but not necessarily a specific one. When multiple equilibria exist, the algorithm may converge to any of them, and our current analysis does not control which one is selected. We will revise the manuscript to clarify this point explicitly and avoid any impression that uniqueness is assumed.
>
> ### On the boundedness of the Bregman divergence and boundary Nash equilibria:
>
> We thank the reviewer for the insightful comment. The reviewer is indeed correct; boundary equilibria are indeed often treated separately in the literature. However, we did not make any assumption that the equilibrium does not live on the boundary; we have to consider the unbounded cases. We will add a discussion of this in the revised manuscript.
>
> ### Choosing p when learners are (partially) oblivious
>
> We want to remark that our framework does not require full knowledge of the game or of other players’ objectives. The regularizer $p_i$ can be chosen individually (and differently) by each player based solely on their own cost structure. Our theorem and analysis are presented with the same $p$ only for a simpler presentation.
>
> ### Regarding realistic choices for $p_i$, a few examples include:
>
> For objectives with known curvature, entropic or Mahalanobis-type regularizers (or quadratic forms adapted to the geometry of $c_i$) may also be suitable.
>
> Even if $c_i$ is not uniformly curved, one can often choose a $p_i$ such that $p_i$ has stronger curvature than $c_i$, ensuring $c_i - \kappa p_i$ is convex for some $\kappa$.
>
>
> ### On the feasibility of $\hat{x}_i^t$​:
>
> We thank the reviewer for pointing this out. The reviewer is correct​, our construction ensures feasibility by design, and we will make this explicit in the updated manuscript.
>
> ### On the relationship with Duvocelle’s results:
>
> We thank the reviewer for pointing this out. We will revise the text to clarify the distinction. Duvocelle’s results apply to strongly monotone games, whereas our results hold for general monotone games. Our results therefore, do not overlap.
>
> ### On the notation $x_i^{t,*}$ and uniqueness (line 300):
>
> We thank the reviewer for the observation. To clarify, $x _i^{t,\ast}​$ refers to one equilibrium point at time, and we do not assume uniqueness of the equilibrium. The analysis holds as long as $x_i^{t,\ast}$​ is any Nash equilibrium, and our results do not require it to be unique. We will revise the notation and clarify this in the text.
>
> ### On the omission of $D_p$​ from the mirror map in Line 303:
>
> Yes, this is deliberate. The result in this section concerns the convergence of the average iterate result for equilibrium tracking, for which we do not need the additional regularization from $D_p$. The inclusion of $D_p$ is crucial for our last-iterate convergence results, but not necessary for average-iterate convergence. We will clarify this distinction in the revision.
>
> ### Line 630-631 on self-concordant barrier with linear term:
>
> We will revise the manuscript and note that $h(x)$ is a self-concordant barrier function. Therefore, we consider a function $f(x) = h(x) + \langle a, x \rangle$, it is still a self-concordant barrier function. This is because the third derivative of any linear function is zero, the addition of the linear term does not affect the third derivative of $f$. Since self-concordance is defined in terms of bounds on the third derivative relative to the second derivative, $f$ inherits the self-concordance of $h$.
>
> ### Line 639, is $p$ defined on $\mathcal{X}_i$ or $\mathcal{X}$.
>
> In our setup, $p$ is defined as a separable function over the combined action space $\mathcal{X} = \prod_{i \in \mathcal{N}} \mathcal{X}i$, meaning $p(x) = \sum{i \in \mathcal{N}} p_i(x_i)$, where each $p_i$ is defined on the individual player’s action space $\mathcal{X}_i$.
>
> Because of this separability, the gradient of $p$ decomposes as, $\nabla p(x) = (\nabla p_1(x_1), \nabla p_2(x_2), \ldots, \nabla p_N(x_N))$, and the Bregman divergence with respect to $p$ satisfies  $D_p(x, x') = \sum _{i \in \mathcal{N}} D _p (x_i, x'_i)$ .
>
> Thus, the inner product difference involving $\nabla p(x) - \nabla p(x')$ naturally decomposes into a sum over the players’ individual spaces, which explains why the inequality $\langle \nabla p(x) - \nabla p(x'), x' - x \rangle \leq - \sum_{i \in \mathcal{N}} ( D_{p_i}(x_i, x'i) + D{p_i}(x'_i, x_i) )$ holds.
>
> We hope that this addresses the reviewer's concerns and we welcome any further questions.

---

> > ### Comment · Reviewer_8yuX · 2025-08-01
> >
> > Thank you for your response.
> >
> > I don't think it was sufficiently clear that p decomposed into p_i throughout.
> >
> > I still think your result is interesting, while also acknowledging the feedback of the other reviewers. I would like to see the limitations of the result discussed in the article (namely that the main result relies on being able to choose a p and that this choice affects the convergence result).
> >
> > Unfortunately I do not feel strongly enough that the article is really strong to argue against the strong feeling of the other reviewers, but I will hold firm on weak accept!

---

> > > ### Author Response · Authors · 2025-08-01
> > >
> > > We thank the reviewer for the comments and support for this paper again.
> > >
> > > We will revise the manuscript to make it clear that $p$ can be decomposed into $p_i$ throughout, and also add a discussion for how $p$ affects the convergence result.

---

### Official Review · Reviewer_KCuo · 2025-06-30

**Clarity:** 2
**Significance:** 2
**Originality:** 2
**Rating:** 4
**Confidence:** 4

**Summary:**

This paper studies the problem of no-regret learning in smooth monotone games with bandit feedback, focusing on developing uncoupled algorithms with non-asymptotic last-iterate convergence rates. The main contribution of the paper is that they proposed a mirror descent-based no-regret online learning algorithm that achieves $O(T^{-1/4})$ last-iterate convergence rate with respect to a problem-dependent Bregman divergence between the iterate and the Nash equilibria. When the loss function is linear (i.e., in two-player zero-sum games), they propose another algorithm with a $O(T^{-1/6})$ last-iterate convergence rate. They also extend the results for time-varying monotone games and present convergence rate results under several bounded variation assumptions.

**Questions:**

1. Does the $O(T^{-1/4})$ convergence rates hold for the gap function or the total gap function? Or in other words, is the $T$-th iterate an  $O(T^{-1/4})$-approximate Nash equilibrium of the game?
2. What is the requirement of choosing $p$ in Algorithm 1? In the current presentation, it seems that one can choose $p$ to be linear (as discussed in the weakness above), but then the guarantee on $D_p(x, x^*)$ becomes vacuous?

**Ethical Concerns:**

["NO or VERY MINOR ethics concerns only"]

**Final Justification:**

I updated my rating to borderline accept, provided that the authors promise to update the current misleading claims (Table 1, abstract, and introduction) on the $T^{-1/4}$ convergence rate on monotone games. It should be made clear that it is a problem-dependent rate that holds for the problem-dependent Bregman divergence and does not imply bounds on the gap functions.

**Limitations:**

Yes.

**Quality:**

2

**Strengths And Weaknesses:**

Strength: This paper tackles an important question on uncoupled learning in smooth monotone games with bandit feedback, for which existing results are limited to either strongly monotone games, zero-sum games, or only asymptotic convergence rates.

Weakness: Although this paper studies an interesting and important question, several of its claims do not seem correct to me, which significantly undermines its contribution. In short, the main result of $O(T^{-1/4})$ last-iterate convergence rates does not hold for the gap function/other standard convergence measures, and the given guarantee could be vacuous. I think a major revision is need to address these issues. Detailed comments below.
1. The claimed $O(T^{-1/4})$ last-iterate convergence rates for general smooth monotone games is not for standard convergence measures such as the gap function, the total gap function, the KL-divergence (or Euclidean distance) between the iterate and a Nash equilibrium. Instead, the proximity to a Nash equilibrium $x*$ is measured by $D_p(x^t, x^{*})$, where $p$ is a **game-dependent** convex function. Specifically, in lines 154-155, the function $p$ is chosen so that $c - \kappa p$ is convex. This is problematic since if we choose $p$ to be a linear function (which is convex and bounded over a compact set), then the corresponding Bregman divergence $D_p(x, x')$ is zero for any two points, making the convergence rate vacuous. Line 193-195 says *"when $c_i(x) = x^2$, $\dots$, we can $\dots$ take $p = x$"*. For this choice of $p$, $D_p(x, x') = 0$ for any two points, and the guarantees in Theorems 5.1, 5.2, 6.1 are vacuous.
2. Continuing on the last point, where the authors give an example where they choose $p = x$ to be a linear function. The sentence on Line 263 *"when $c_i$ is linear, there does exist a $p$ that is convex while making $c_i - \kappa p$ convex."* makes no sense since we can choose $p$ to be a linear function and the resulting function $c_i - \kappa p$ is linear, thus convex.
3. The choice of $p$ also gives rise to a minor concern on the uncoupledness of the algorithm. Typically, when the players are playing an unknown game, it is common to assume they know some (upper bounds of) game parameters, such as the smoothness. For example, as long as the agents know they are playing a $L$-smooth monotone game (the specific game instance remains unknown), they can choose their step size and achieve convergence in any $L$-smooth monotone game. But to implement Algorithm 1 in the paper, the agents must also agree on a function $p$ which depends on the unknown game instance, and the choice of $p$ must be different even in the class of $L$-smooth monotone games (and even if they know $L$). This issue affects the uncoupledness of the algorithm.

---

> ### Author Rebuttal · Authors · 2025-07-30
>
> We thank the reviewer for their detailed and thoughtful comments. We appreciate the recognition of the importance of the problem we study and would like to clarify a few key points.
>
> ### On the concern regarding the use of Bregman divergence and the possibility of vacuous bounds
>
> We agree that if $p$ is chosen to be a linear function, the corresponding Bregman divergence becomes zero, and the bound would indeed be vacuous. However, this does not affect the validity of our result, $p$ is a design choice, and our framework allows for choosing $p$ to be sufficiently curved (i.e., not flat), ensuring the Bregman divergence is meaningful. We will clarify this point in the revision to avoid any misunderstanding.
>
> ### On the issue of uncoupledness and agreement on $p$
> It is not necessary for all players to agree on the same function $p$. Our result holds even when each player selects their own $p_i$, as long as the key assumption, namely, that $c_i−\kappa pi$ is convex, is satisfied. In the case where all players use the same $p$, we can express convergence using the total Bregman divergence $\sum_i D_p(x_i, x_i^\ast)$. If players use different pip_ipi​, the result generalizes naturally to $\sum_i D_{p_i} (x_i, x_i^\ast)$. We assume that players take the same $p$ for a simpler presentation.
> We will revise the manuscript to make these points more explicit and to address the valid concerns raised. Thank you again for the valuable feedback.

---

> > ### Comment · Reviewer_KCuo · 2025-08-01
> >
> > I thank the authors for the response. After reading other reviewers' responses, I still have concerns on the results and presentation.
> >
> > (1) Does the $O(T\^{-1/4})$ convergence rate hold for the gap or the total gap function? Or, in other words, **is the
> > $T$-th iterate an $O(T\^{-1/4})$-approximate Nash equilibrium of the game?**
> >
> > This was one of my initial questions and remains unanswered. If the $T$-th iterate is not an $O(T^{-1/4})$-approximate Nash equilibrium of the game, this should be made explicit. In that case, the new convergence rates are with respect to a non-standard measure that does not imply bounds on the standard gap function. Table 1 and the abstract should be revised to reflect this distinction clearly.
> >
> > At present, most of the existing results reported in Table 1 hold for the standard gap function and thus imply approximate Nash equilibria. Without clarification, the current presentation gives the misleading impression that the new results also apply to the gap function. I believe Reviewer G5Qr may have been confused by this lack of precision.
> >
> > (2) Regarding vacuous bounds. If the $O(T^{-1/4})$ rate concerning a Bregman divergence does not hold for monotone games with linear utility, this limitation should also be clearly stated in the abstract and Table 1 (currently the presentation is misleading). A brief discussion of this restriction would be helpful, especially since prior results for monotone games do not have this caveat.

---

> ### Author Response · Authors · 2025-08-01
>
> We thank the reviewer again for the helpful comments and suggestions.
>
> We will revise Table 1 to include a column specifying the convergence metrics used. Our results are stated in terms of $D_p(x_t, x^\ast)$, which, under Assumption 3.1, can implies that $x_t$ is an $\epsilon$-Nash equilibrium. The result for the linear zero-sum case is stated in terms of the total gap function, which similarly implies an $\epsilon$-NE. Therefore, even though some prior works use the (total) gap function while we use $D_p$, the table remains a meaningful basis for comparison. That said, we do not claim that $D_p$ and the (total) gap function are directly comparable, nor that one necessarily upper bounds the other.
>
> We will add another paragraph following the presentation of Table 1 to explain the differences in convergence metrics and the limitation of $D_p$.
>
> We hope this address the reviewer's concern, and we welcome any further questions.

---

### Official Review · Reviewer_G5Qr · 2025-07-01

**Clarity:** 3
**Significance:** 3
**Originality:** 2
**Rating:** 4
**Confidence:** 3

**Summary:**

The paper studies the problem of no-regret learning in general monotone problems under bandit feedback. The main result is a mirror descent-based algorithm that attains faster rates in strongly monotone and monotone problems. The results are also extended to time-varying problems, improving again over the existing rates. Experiments are also provided to support some of the theoretical findings.

**Questions:**

I have two main questions:

- Can the authors explain the incongruity highlighted above related to monotone problems and zero-sum games? Is it the case that the result for monotone problems requires stronger assumptions?
- Can the authors provide the precise statements for the theorems? The key assumptions were not clear even after reading the appendix.

**Ethical Concerns:**

["NO or VERY MINOR ethics concerns only"]

**Final Justification:**

In my original review I was confused about a certain inconsistency in one of the main results of the paper. The authors were able to address that point in the rebuttal; even though Table 1 contained a really confusing claim, that issue can be easily corrected for the final version. Overall, I believe the result is significant enough for publication.

**Limitations:**

Yes.

**Paper Formatting Concerns:**

No paper formatting concerns.

**Quality:**

2

**Strengths And Weaknesses:**

On the positive side, the paper studies a well-motivated and important problem. While there has been much work studying last-iterate convergence in games, most existing results apply under full feedback. This paper is part of a recent line of work that tries to extend those results to the bandit feedback setting. There are many challenges in extending those results to the bandit feedback setting. To my knowledge, the results reported by the paper provide significant improvements over the current state of the art. Given the significance of this problem, I believe this is a valuable contribution. The technical approach is also non-trivial and contains some interesting ideas. Although the main ideas appear to be known from prior work, I believe that the way those prior techniques are used in combination in this setting is new. Moreover, the writing overall is reasonable and the paper goes a good job at motivating the key ideas.

On the other hand, I have some concerns about the paper. First, there is a claim that I find very confusing. Table 1 claims that the rate obtained for zero-sum games is worse than general monotone problems. This doesn't make sense since (two-player) zero-sum games are monotone. The only way to reconcile this is that the main result for the monotone case requires further assumptions that are hidden. It is essential that the authors can address this confusion. Also, the statements of the theorems are not rigorous enough. They do not clearly state the main assumptions and most of the parameters that appear in the statements are not defined or specified in that context. I expect to see formal, self-contained statements.

---

> ### Author Rebuttal · Authors · 2025-07-30
>
> We thank the reviewer for the detailed and constructive comments. We hope the following clarify the reviewer's concern.
>
> ### Regarding the confusion in Table 1
>
> We agree that zero-sum games are a subclass of monotone problems, and we apologize for the lack of clarity. The linear zero-sum case does not require stronger assumptions. However, due to the flatness of the cost functions in this setting, our analysis framework requires special treatment. In particular, the Bregman divergence induced by the regularizer ppp becomes ill-defined, which prevents a direct comparison using that measure.
> Instead, our results for the linear zero-sum case are stated in terms of the standard gap function, $\sum_i <\nabla_i c(x_i), x_i - x_i^\ast >$, which remains meaningful in this context. While this differs in form from the result in the general monotone setting, it is not weaker.
> We will revise the presentation of Table 1 and the accompanying discussion to clarify this distinction. We will also provide formal, self-contained theorem statements in the main text, including all assumptions and parameter definitions.
>
> ### Can the authors provide the precise statements for the theorems? The key assumptions were not clear even after reading the appendix.
>
> The only parameters required in our main theorem are $\eta_t$ and $\delta_t$, both of which are stated explicitly in the theorem. We do not rely on any additional hidden assumptions; everything is captured by Assumption 3.1. To improve clarity, we will revise the theorem statement to directly reference Assumption 3.1 and ensure that all relevant parameters and conditions are clearly defined and self-contained.

---

> > ### Comment · Reviewer_G5Qr · 2025-08-01
> >
> > I thank the authors for the response. I am afraid I don't quite follow their response concerning the confusion in Table 1. Isn't the result for linear zero-sum weaker than the general monotone setting? Is there some statement in Table 1 that is currently wrong? How exactly are you planning to revise Table 1?

---

> > > ### Author Response · Authors · 2025-08-01
> > >
> > > We thank the reviewer again for the questions. The result for linear zero sum is not weaker because it is an upper bound for $\sum_i \langle \nabla_i c(x_i), x_i - x_i^\ast \rangle$ rather than the $D_p(x_t, x^\ast)$. Since these two results bound fundamentally different quantities, it is not meaningful to compare them directly in terms of strength.
> > >
> > > The reason why we make the linear zero-sum a special case and provide an upper bound for $\sum_i \langle \nabla_i c(x_i), x_i - x_i^\ast \rangle$ is that $D_p(\cdot, \cdot)$ is meaningless when $p$ is linear. We also want to know that this $\sum_i \langle \nabla_i c(x_i), x_i - x_i^\ast \rangle$ is also the common convergence metrics used in previous works [Cai et al. 2023], which then allows for fair and direct comparisons to existing results.
> > >
> > > Although Table 1 is not wrong in terms of the claims, we agree that Table 1 may be confusing as it does not clearly distinguish the difference in convergence metrics. To address this, we plan to create a separate table specifically for the linear zero-sum case, explicitly highlighting the differing convergence criteria.

---

> > > > ### Comment · Reviewer_G5Qr · 2025-08-08
> > > >
> > > > I thank the authors for the clarification. I have increased my score. Please make sure to appropriately revise Table 1 and the introduction to make clear the convergence metric.

---

### Official Review · Reviewer_X31B · 2025-07-03

**Clarity:** 3
**Significance:** 4
**Originality:** 3
**Rating:** 5
**Confidence:** 3

**Summary:**

The paper considers monotone games in a setting where the players
operate independently (uncoupled), and observe their costs/payoffs but
not the gradient of the cost function (bandit feedback). The class of
monotone games include convex-concave games, Cournot competition, and
splittable routing games.

In particular, they analyse the last-iterate distance to the Nash
equilibrium of an algorithm based on mirror descent. Given that only
bandit feedback is available, the algorithm uses an ellipsoidal
gradient estimator.

In overall, they prove convergence O(1/T^1/2) for strongly monotone
games, O(1/T^1/4) for monotone games, and O(1/T^1/6) for linear games
(results are also available for time-varying games). According to the
authors, this is the first last-iterate convergence results available
for monotone games. Previous results have either been asymptotic, or
have made stronger assumptions about the games.

There is also an experimental comparison between their algorithm and
OMD and GD.

**Questions:**

To follow the analysis, it would be helpful if the paper provides a
very brief introduction to mirror descent.

Algorithm 1, line 4. You could state that z_i^t is sampled u.a.r. from
the d-Sphere, and before receiving the bandit feedback.

Algorithm 1, line 6. It would be useful with a comment about the
hardness of solving the optimisation problem in Eq (1).

As far as I can see, there is a typo in the statement of Lemma
J.2. Instead of an identity, the result should be an upper bound. In
particular, the Holder inequality is stated as an equality. This
change would not have any impact on the overall result since Lemma J.2
is always applied as an upper bound.

**Ethical Concerns:**

["NO or VERY MINOR ethics concerns only"]

**Final Justification:**

Based on the discussions after the rebuttal and taking into my own comments regarding some weaknesses in the experimental evaluation, I felt that a score of 6 (flawless) was too high. I have therefore reduced the score to 5.

**Limitations:**

Yes

**Quality:**

4

**Strengths And Weaknesses:**

In overall, this is a highly significant contribution that deserves to
be published.

I must admit that the paper is outside my core area of expertise. I
have therefore not had the opportunity to read every proof in detail.
However, I have some minor remarks below.

The experimental study is rather light touch, and only uses 5
different random seeds. The authors should consider more independent
runs to take into account variance. It would also be interesting to
see how the behaviour of the algorithm scales with increasing
dimension d (currently only very low dimensions are explored). The
experimental results should be discussed more in-depth and contrasted
with the theoretical results.

---

> ### Author Rebuttal · Authors · 2025-07-30
>
> We thank the reviewer for the encouraging feedback. We're glad the contribution is viewed as significant.
> We will revise the manuscript accordingly. Specifically, we will:
>
> 1. Increase the number of random seeds and include experiments in higher dimensions to better assess variance and scalability.
>
>
> 2. Expand the discussion of experimental results and relate them more clearly to the theoretical findings.
>
>
> 3. Add a brief introduction to mirror descent for clarity.
>
>
> 4. Clarify in Algorithm 1 that zitz_i^tzit​ is sampled uniformly at random from the ddd-sphere, before receiving feedback.
>
>
> 5. Add a remark on the hardness of solving Eq. (1).
>
>
> 6. Fix the typo in Lemma J.2 and restate it as an upper bound.

---

### Note · Authors · 2025-08-12

We sincerely thank the reviewers for their constructive feedback and the opportunity to clarify our contributions. The main concerns raised focus on clarifying the convergence metrics employed in our results, specifically on comparing the convergence measured by the Bregman divergence from that measured by the total gap function, and on explicitly addressing how the choice of regularizer $p$ influences the convergence guarantees. We have clarified the concerns raised during the rebuttal. In the revised manuscript, we will make the following revisions.

1. Table 1 will include a new column specifying the convergence measure used in each work. We will explicitly state that our results are mostly in terms of the Bregman divergence, which under Assumption 3.1 may imply an $\epsilon$-Nash equilibrium, and, for the linear zero-sum case, in terms of the total gap function, which also implies an $\epsilon$-NE. While these measures are meaningful for comparison, we will make clear that they may not be directly comparable to the (total) gap function used in some prior works. Specifically, neither of the two measures upper bounds the other. The abstract and the discussion following Table 1 will also be revised accordingly to avoid any possible confusion.
2. We will also make explicit that the choice of regularizer $p$ (and its decomposition into $p_i$) affects the convergence result, and discuss the implications of this choice. We will remark that it is not necessary for all players to agree on the same function $p$, and therefore our algorithm is uncoupled.

We appreciate the reviewers’ recognition of the interest in our results, and we believe these revisions will address the concerns raised while making the presentation and implications of our work more precise.

Authors of Paper #18394

---

### Decision · Program_Chairs · 2025-09-17

**Decision:**

Accept (poster)

**Comment:**

This paper studies last-iterate convergence in monotone games with bandit feedback via mirror descent, achieving improved rates in several settings. Reviewers agree that the problem is important and interesting, though concerns arose about clarity of convergence measures and the algorithm's strong dependence on the knowledge of parameter $p$ of other players, and these are never clarified in the Introduction section and the comparison table. The authors promised to clarifying things in the revised version. After discussion, reviewers converged to borderline/weak accept, with no strong objections. Overall, the work is technically solid and merits acceptance pending the promised revisions.